

# The complete chloroplast genome of *Dendrobium nobile*, an endangered medicinal orchid from north-east India and its comparison with related *Dendrobium* species

Ruchishree Konhar[1,2,3], Manish Debnath[1], Santosh Vishwakarma[1], Atanu Bhattacharjee[1], Durai Sundar[4], Pramod Tandon[5], Debasis Dash[2,3] and Devendra Kumar Biswal[1]

[1] Bioinformatics Centre, North-Eastern Hill University, Shillong, Meghalaya, India
[2] Informatics and Big Data, CSIR-Institute of Genomics and Integrative Biology, New Delhi, India
[3] Academy of Scientific and Innovative Research, Ghaziabad, Uttar Pradesh, India
[4] Department of Biochemical Engineering and Biotechnology, Indian Institute of Technology Delhi, New Delhi, India
[5] Biotech Park, Kursi road, Lucknow, Uttar Pradesh, India

## ABSTRACT

The medicinal orchid genus *Dendrobium* belonging to the Orchidaceae family is a huge genus comprising about 800–1,500 species. To better illustrate the species status in the genus *Dendrobium*, a comparative analysis of 33 available chloroplast genomes retrieved from NCBI RefSeq database was compared with that of the first complete chloroplast genome of *D. nobile* from north-east India based on next-generation sequencing methods (Illumina HiSeq 2500-PE150). Our results provide comparative chloroplast genomic information for taxonomical identification, alignment-free phylogenomic inference and other statistical features of *Dendrobium* plastomes, which can also provide valuable information on their mutational events and sequence divergence.

Corresponding authors
Debasis Dash, ddash@igib.res.in
Devendra Kumar Biswal, devbioinfo@gmail.com

## INTRODUCTION

*Dendrobium* is a huge genus of the tribe Dendrobieae (Orchidaceae: Epidendroideae) that was established by Olof Swartz in 1799. It includes approximately 800–1,500 species and occurs in diverse habitats throughout much of Southeast Asia, including China, Japan, India, and the Philippines, Indonesia, New Guinea, Vietnam, Australia and many of the islands in the Pacific (*Wood, 2006*).

Many species and cultivars of this genus are well-known floral motifs and have featured in artwork. *Dendrobium* orchids are popular not only for their visual appeal in cut flower market, but also for their herbal medicinal history of about 2,000 years in east and south Asian countries (*Bulpitt et al., 2007*). Many species in this genus have been extensively used as herbal medicines for several hundreds of years in treating diseases like kidney and lung

ailments, gastrointestinal problems, lumbago and arthralgia. The plant extracts are also used as tonic for strengthening body's immunity and improving sexual potency. However, many *Dendrobium* species in the wild face an extreme threat of extinction due to their low germination and slow growth rate, habitat decline and over exploitation arising out of anthropogenic activities (*Kong et al., 2003*).

*Dendrobium* orchids have overwhelmed researchers because of their high economic importance in global horticultural trade and in Asian traditional medicine leading to extensive plant systemic studies particularly in species identification, novel marker development, breeding and conservation. In the past two decades, promising advances have been made in areas of molecular taxonomy, plant systematics and selective breeding of *Dendrobium* species by intensive use of molecular markers. Recently, a variety of molecular markers like microsatellite (SSR), Random Amplified Polymorphic DNA (RAPD) and Amplified Fragment Length Polymorphism (AFLP) markers including several other DNA barcode markers from different loci of nuclear and chloroplast (cp) regions have been developed to study *Dendrobium* diversity. However, these species are notoriously difficult to identify (*Teixeira da Silva et al., 2016*).

The complete chloroplast genome usually contains a uniparentally inherited DNA, a feature which makes it an obvious choice for plant taxonomical analyses, phylogenomics and phylogeographic inferences at different taxonomic levels. One such classic example is the study of phylogenetic relationships among all families in the Order Liliales, based on 75 plastid genes from 35 species in 29 genera and 100 species spanning all monocot and major eudicot lineages, where underlying results were calibrated against 17 fossil dates to redefine the monocot evolutionary timelines (*Givnish et al., 2016b*). The significance of plastome-scale data was very well demonstrated in another study that highlighted a new functional model for understanding monocot evolution and some of their derived morphological features by way of convergent evolution from submersed aquatic ancestors (aquatic Hydatellaceae) (*Givnish et al., 2018*). The evolution of orchids, the largest and most diverse family of flowering plants second only to Asteraceae on Earth has long puzzled Charles Darwin and many other scientists. Recent advances in chloroplast genomics are giving researchers insights into the evolutionary history of these plants. One such study hypothesizes orchids to have arisen in Australia 112 Ma followed by migration to the Neotropics via Antarctica by 90 Ma. With the use of a combination of plastid genes, it was established that orchids and epidendroids exhibited maximally accelerated net diversification in Southeast Asia and the Neotropics respectively (*Givnish et al., 2016a*).

Studies pertaining to plastome genome sequences are useful in investigating the maternal inheritance in plants, especially those with polyploid species, owing to their high gene content and conserved genome structure (*Birky, 1995*; *Soltis & Soltis, 2000*; *Song et al., 2002*). Many species of orchids and other flowering plants exhibit rapid evolution and diversity. One of the main reasons for such diversity can be attributed to allopolyploidy or genetic redundancy, in which there are more than one gene involved in performing a particular task. In cases of useful mutation, plants evolve into new species. Hybridization and polyploidy are the decisive forces behind evolution and speciation. In the past there

have been studies where a combination of AFLPs, cpDNA markers and flow cytometry was harnessed to investigate the evolutionary outcomes of hybridization between two endemic Ecuadorian species of Epidendrum (Orchidaceae) in three hybrid zones. The outcome of this study highlights the importance of hidden hybrid genotypes and their frequency which could help unravel the mysteries behind orchid evolution (*Marques et al., 2014*). The advent of high-throughput sequencing technologies has enabled a rapid increase in the rate of completion of cp genomes with faster and cheaper methods to sequence organellar genomes (*Saski et al., 2007*; *Cronn et al., 2008*). At the time of writing this manuscript, cp genomes from 33 *Dendrobium* species have been reported as per NCBI Organellar genome records (https://www.ncbi.nlm.nih.gov/genome/browse#!/organelles/dendrobium).

*D. nobile* Lindl. is one of the many highly prized medicinal plants in the genus *Dendrobium*. It is an endangered medicinal orchid listed in the Convention on International Trade in Endangered Species of Wild Fauna and Flora (CITES) Appendix II that demands immediate attention for its protection and propagation. Here, we report the first complete cp genome of *D. nobile* from north-east India based on next-generation sequencing methods (Illumina HiSeq 2500-PE150) and further compare its structure, gene arrangement and microsatellite repeats with 33 existing cp genomes of *Dendrobium* species. Our results provide comparative chloroplast genomic information for taxonomical identification, phylogenomic inference and other statistical features of *Dendrobium* plastomes. These can give further insights into their mutational events and sequence divergence. The availability of complete cp genome sequences of these species in the genus *Dendrobium* will benefit future phylogenetic analyses and aid in germplasm utilization of these plants.

# MATERIALS AND METHODS

## Sample collection, DNA extraction and sequencing

Fresh leaves of *D. nobile* were collected from plants growing in greenhouses of National Research Centre for Orchids, Sikkim, India and voucher specimen was deposited in Botanical Survey of India as well as in the Department of Botany, North-Eastern Hill University, Shillong. The high molecular weight cpDNA was extracted using a modified CTAB buffer, and treated according to a standard procedure for next generation sequencing on Illumina HiSeq 2500-PE150. The quality and quantity of the genomic DNA was assessed through agarose gel electrophoresis, Nanodrop and Qubit detection method. The experiments included both paired-end and mate-pair libraries. Tagmentation was carried out with ~4 µg of Qubit quantified DNA and the tagmented sample was washed using AMPURE XP beads (Beckman Coulter #A63881) and further exposed to strand displacement. The strand-displaced sample of 2–5 kb and 8–13 kb gel was size selected and taken for overnight circularization. The linear DNA was digested using DNA Exonuclease. Further the circularized DNA molecules were sheared using Covaris microTUBE, S220 system (Covaris, Inc., Woburn, MA, USA) for obtaining fragments in the range 300 to 1,000 bp. M280 Streptavidin beads (ThermoFisher Scientific, Waltham, MA) was used to cleanse the sheared DNA fragments with biotinylated junction adapters. The bead-DNA complex was subjected to End Repair, A-Tailing and Adapter ligations.

## Data processing

The data quality assessment for Illumina WGS raw reads was carried out using FastQC tool. Perl scripts were written for adapter clipping and low quality filtering. Chloroplast genomes of *D. officinale*, *D. huoshanense* and *D. strongylanthum* retrieved from NCBI-RefSeq database was used as reference for the assembly. BWA-MEM algorithm with default parameter settings was used for aligning the adapter clipped and low quality trimmed processed reads with the *Dendrobium* cp genomes (*Li & Durbin, 2009*). SPAdes-3.6.0 program was used for k-mer based (k-mer used 21, 33, 55 and 77) de-novo assembly with the aligned reads and the quality of the assembled genome was gauged using Samtools and Bcftools (read alignment and genome coverage calculation) (*Bankevich et al., 2012*) (https://samtools.github.io/bcftools/bcftools.html). The cp genome of *D. nobile* was also generated through reference-assisted assembly using the high quality paired-end libraries by NOVOPlasty (*Dierckxsens, Mardulyn & Smits, 2017*) for further validation. It is specifically designed for de novo assembly of mitochondrial and chloroplast genomes from WGS data with the aid of a reference or seed sequence. The seed sequence can correspond to partial or complete sequence of chloroplasts of closely to distantly related species. The cpDNA RefSeq sequence of *D. officinale* was used as a seed sequence to perform reference-assisted assembly.

## Genome annotation and codon usage

Basic Local Alignment Search Tool (BLAST; BLASTN, PHI-BLAST and BLASTX) (*Altschul et al., 1997*), chloroplast genome analysis platform (CGAP) (*Cheng et al., 2013*) and Dual Organellar GenoMe Annotator (DOGMA) (*Wyman, Jansen & Boore, 2004*) was used to annotate protein-coding and ribosomal RNA genes. The boundaries of each annotated gene with putative start, stop, and intron positions were manually determined by comparison with homologous genes from other orchid cp genomes. Further tRNA genes were predicted using tRNAscan-SE (*Lowe & Eddy, 1997*) and ARAGORN (*Laslett & Canback, 2004*). RNA editing sites in the protein-coding genes (PCG) of *D. nobile* were predicted using Plant RNA Editing Prediction & Analysis Computer Tool (PREPACT) (http://www.prepact.de). For this analysis, *D. nobile* cp genome was BLAST aligned against *Nicotiana tabacum*, *Oryza sativa Japonica* Group, *Phalaenopsis aphrodite* subsp. *Formosana*, *Physcomitrella patens* subsp. *patens* and *Zea mays* with a cutoff *E*-value set to 0.08. The circular genome map was drawn in OrganellarGenomeDRAW (*Lohse et al., 2013*) followed by manual modification. The sequencing data and gene annotation were submitted to GenBank with accession number KX377961. MEGA 7 was used to analyze and calculate GC content, codon usage, nucleotide sequence statistics and relative synonymous codon usage (RSCU) (*Kumar, Stecher & Tamura, 2016*).

## Gene Ontology annotation and assignment of GO IDs

Gene Ontology (GO) annotation of *D. nobile* chloroplast genes was carried out in Blast2GO (*Conesa et al., 2005*) by blast aligning the gene sequences from the GenBank annotation files to Orchidaceae sequences in non-redundant (nr) database with an e-value cutoff of $1e^{-5}$ and queried in InterProScan (*Jones et al., 2014*). GO mapping and annotation

of genes followed this from blast results and were subsequently merged with GO IDs from InterProScan. The merged GO annotations were validated based on True-Path-Rule by removing redundant child terms for each gene sequence. The GO annotations were slimmed down using plant-slim option.

## Simple sequence repeats analysis

MISA (http://pgrc.ipk-gatersleben.de/misa/misa.html), a tool for identification and location of perfect microsatellites and compound microsatellites was used to search for potential simple sequence repeats (SSRs) loci in the cp genomes of different *Dendrobium* species. The threshold point for SSRs identification was set to 10, 5, 4, 3, and 3 for mono-, di-, tri-, tetra-, and penta-nucleotides SSRs, respectively. All the repeats found were manually curated and the redundant ones were removed.

## Phylogenetic reconstruction with whole genome alignment and rearrangement analysis

For phylogenetic reconstruction, we included *D. nobile* cp genomes from India and China along with 32 other *Dendrobium* cp genomes retrieved from GenBank. Four *Goodyera* species were taken as outgroup. The cp genome sequences were aligned with MAFFT v7.0.0 (*Katoh & Standley, 2013*) and manually curated by visual inspection. PCGs as well as whole cp genomes were used for Bayesian phylogenetic reconstruction using MRBAYES 3.2.6 (*Huelsenbeck & Ronquist, 2001*). To further validate our results we employed "k-mer Based Tree Construction" in CLC Genomics Workbench that uses single sequences or sequence lists as input and creates a distance-based phylogenetic tree. For visualization and testing the presence of genome rearrangement and inversions, gene synteny was performed using MAUVE as implemented in DNASTAR 12.3 with default settings. Comparative analysis of intra nucleotide diversity (*Pi*) within the *Dendrobium* cp genomes was performed using MEGA 7.

## Single nucleotide polymorphism identification and phylogenetic analysis without genome alignment

Phylogenetic tree was constructed based on the Single Nucleotide Polymorphisms (SNPs) identified in the whole cp genomes using kSNP3.0 with default settings except for k-mer size (*Gardner, Slezak & Hall, 2015*). SNPs were identified with k-mer size set to 23, based on which, approximately 79% of the k-mers generated from median-length genome were unique.

# RESULTS

## Genome organization and features

The complete cp genome of *D. nobile* was determined from the data generated out of a whole genome project initiative of the same species by Paired-end and Mate pair data from Illumina HighSeq with 150*2 and Illumina NextSeq500 with 75*2 respectively. Further the aligned Illumina reads were separated and assembled using CLC Main Workbench Version 7.7.1 into the single longest scaffold. The *D. nobile* cp genome is a typical circular double-stranded DNA with a quadripartite structure; it is 152,018 bp in size and consists of

Large Single Copy (LSC) (1..84,944; 84,944 bp), Small Single Copy (SSC) (111,230..125,733; 14,504 bp), and two Inverted Repeat (IR) regions of 26,285 bp: IRA (84,945..111,229) and IRB (125,734..152018). In total 134 unique genes (79 PCGs, 8 rRNA genes, 7 pseudogenes and 38 tRNA genes) were successfully annotated, of which 12 genes {rps16, atpF, rpoC1, ycf3, rps12 (2), clpP, petB, rpl2 (2), ndhB (2)} are reported with introns (Fig. 1). We could identify a total of 20, 81 and 11 genes duplicated in the IR, LSC and SSC regions respectively in the *D. nobile* cp genome. There were a total of 49 RNA editing sites predicted in 23 genes of *D. nobile* cp genome. The whole chloroplast genome alignment included 34 *Dendrobium* species and four species from the genus *Goodyera* as outgroup. Each genome's panel contained its name, sequence coordinates and a black coloured horizontal centre line with coloured block outlines appearing above and below it. Homology between the cp genomes is represented by each block with the genes, internally free from genomic rearrangement, connected by thin lines to similarly coloured blocks depicting comparative homology between the genomes (Fig. 2). The positions of LSC/IRA/SSC/IRB borders revealed similar structures at the IR/LSC junction in the overall alignment of *Dendrobium* whole cp genomes (Fig. 3).

## Gene ontology mapping and annotation

We further analyzed the *D. nobile* coding cp genome sequences using the Blast2GO suite and annotated the sequences for three GO terms (biological process, molecular function, and cellular component). In case of GO term there were a total of 231 annotations in biological process (P), molecular function (F) and cellular compartment (C) level. In the category of biological processes a large number of these sequences are annotated for translation, photosynthesis, metabolic processes, and ribosome biogenesis. Similarly, for the GO term molecular function, the top GO categories include functions related to structural molecule activity, catalytic activity, ion and rRNA binding, transporter and transferase activity. Finally, terms including membrane, ribosome and thylakoid were annotated GO categories for cellular compartment with most of the sequences. These results are summarized along with the information on RNA editing sites in Table 1.

## Simple sequence repeat identification

SSRs were identified in MISA perl scripts with a minimum of 10 bp repeats among all the *Dendrobium* species. Of all the SSRs, the mononucleotide A/T repeat units occupied the highest proportion. A higher proportion of di-, tri- repeats are reported rather than tetra- and penta-nucleotide repeats across *Dendrobium* cp genomes (Fig. 4).

## Phylogenetic analysis

Phylogenetic analyses of chloroplast PCGs from *Dendrobium* species were performed with or without partitions of sequences. Both Bayesian and K-mer based trees (Figs. 5 and 6) recovered a monophyly of the *Dendrobium* species, irrespective of whether or not the partitions of sequences were incorporated in the analysis supported by strong bootstrap values. The phylogenetic analyses based on complete cp genomes, suggested that five major subgroups within the genus *Dendrobium* evolved in a nested evolutionary relationship. *D. aphyllum, D. parishii, D. loddigesii* and *D. primulinum* are the most recently evolved

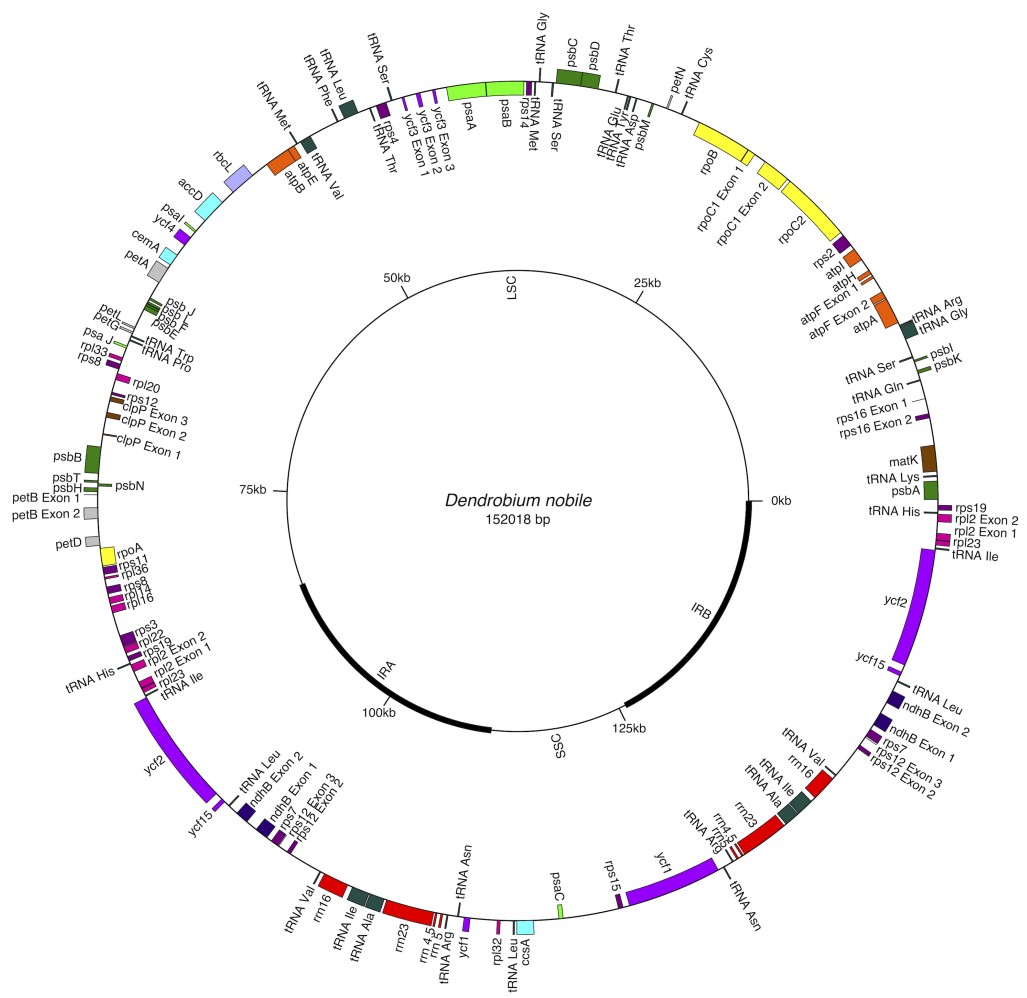

**Figure 1  Gene map of *Dendrobium nobile* chloroplast genome from north-east India.** Genes shown inside the circle are transcribed clockwise, and those outside are transcribed anticlockwise. Color coding indicates genes of different functional groups. A pair of inverted repeats (IRA and IRB) separate the genome into LSC and SSC regions.

species that nested into a single monophyletic sub group within the *Dendrobium* clade. *D. chrysotoxum* and *D. salaccense* were a bit primitive on the evolutionary ladder in the phylogenetic tree. *Goodyera* species emerged as the outgroup that claded separately in the over all tree topology. Similar results were also obtained in the alignment free phylogenetic tree with SNPs (Fig. 6).

## DISCUSSION

### Potential RNA editing sites

RNA editing is involved in plastid posttranscriptional regulation and thus provides an effective way to create transcript and protein diversity (*Chen & Bundschuh, 2012*; *Knoop, 2011*). In Orchidaceae, RNA editing sites were identified in 24 protein-coding transcripts

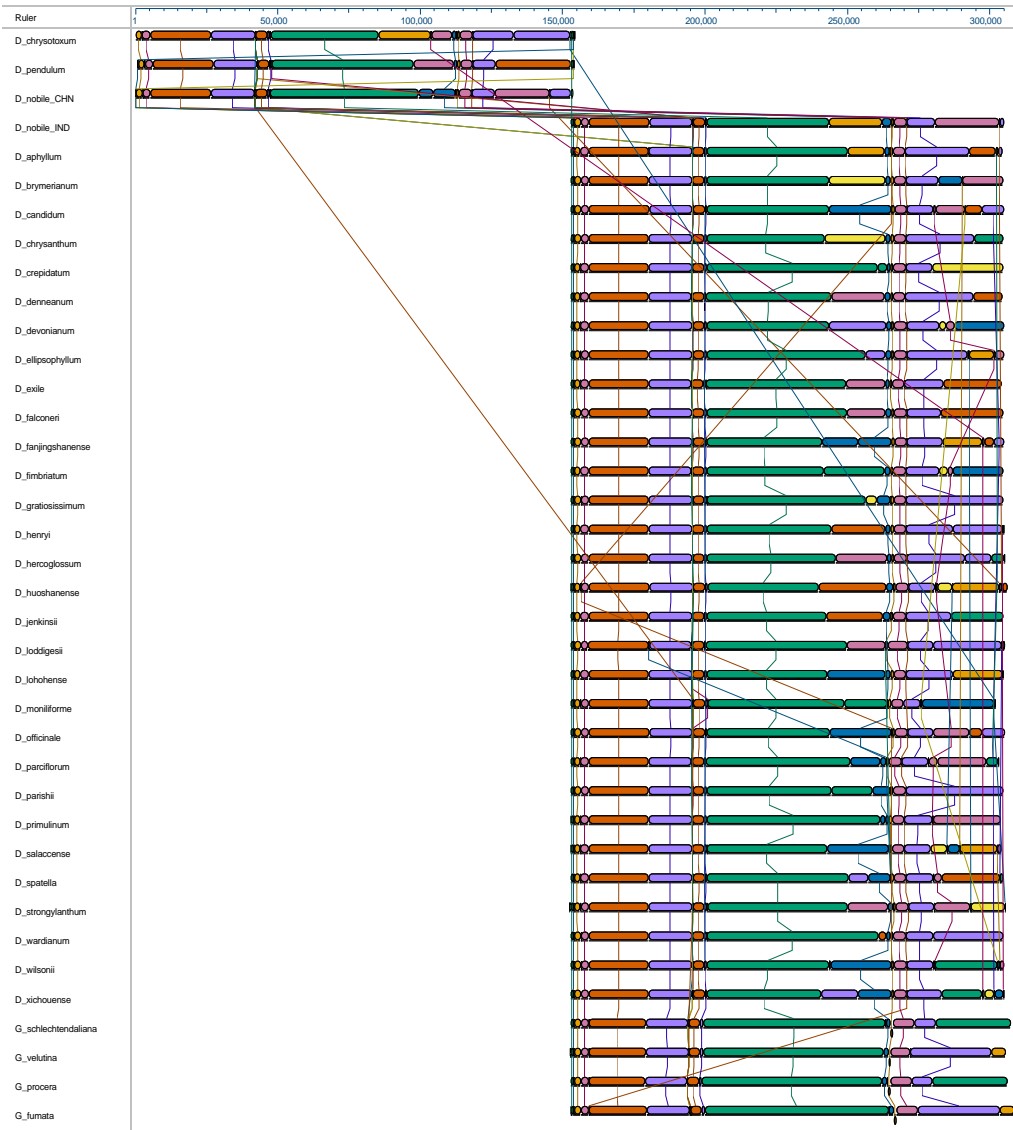

**Figure 2   Whole chloroplast genome alignment of 38 orchid species.** The whole chloroplast genome alignment includes 34 *Dendrobium* species and four species from the genus *Goodyera* as outgroup. Each genome's panel contains its name, sequence coordinates and a black coloured horizontal centre line with coloured block outlines appearing above and below it. Each block represents homology with the genes, internally free from genomic rearrangement, connected by lines to similarly coloured blocks depicting comparative homology across genomes.

in *P. aphrodite* (*Zeng, Liao & Chang, 2007*). Earlier studies indicate RNA editing sites from the same subfamily to be more conserved than those from different subfamily (*Luo et al., 2014*). However, orchids and other angiosperms have relatively less common editing sites. For example, orchids and *Cocos nucifera* share 10 potential RNA editing sites; comparisons among *Nicotiana tabacum*, *Arabidopsis thaliana* and orchid RNA editing sites have shown low conservation of editing sites (one common editing site in *rpo* B) (*Luo et al., 2014*).

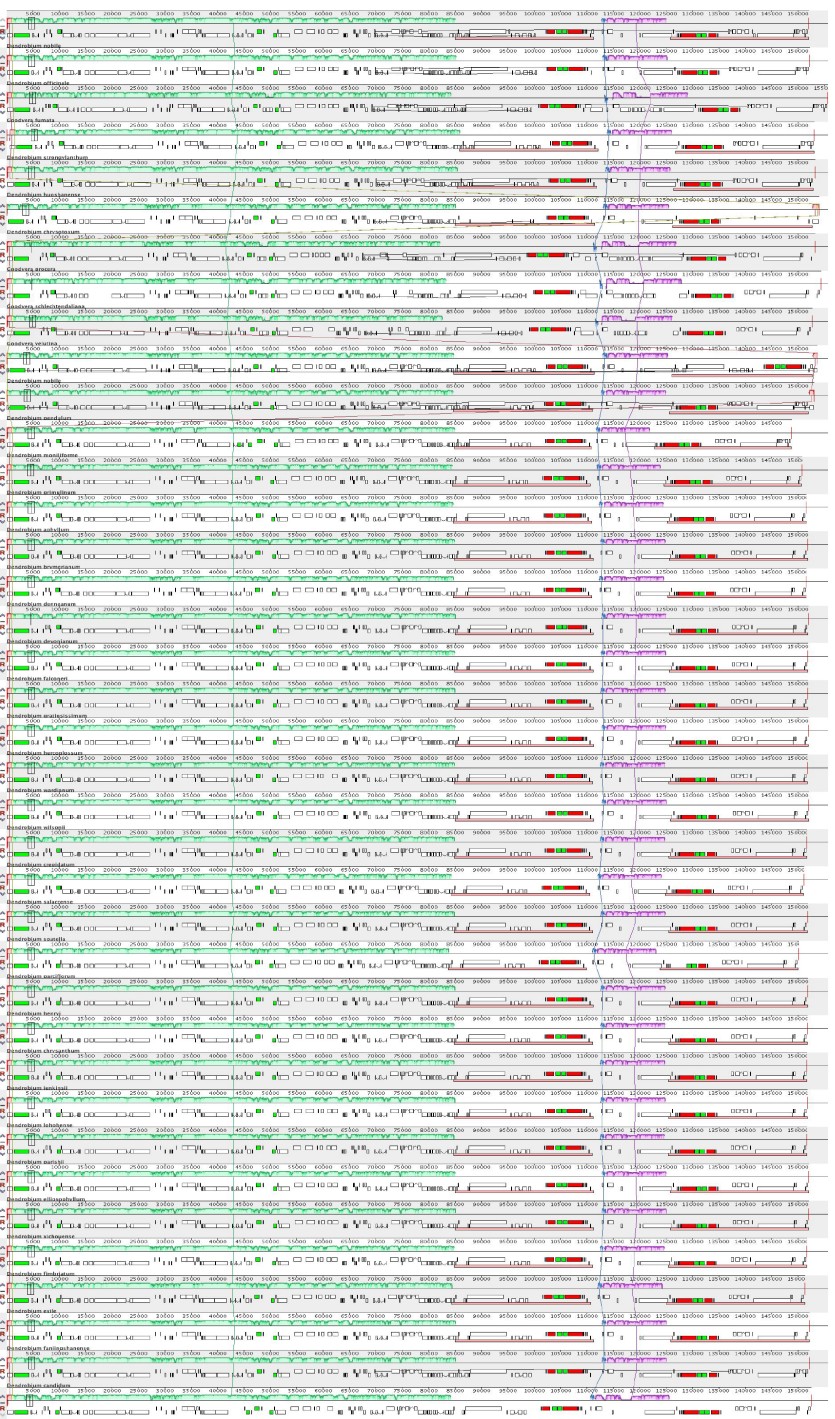

**Figure 3  Comparison of the borders of LSC, SSC and IR regions across *Dendrobium* chloroplast genomes.**

Our studies congruently predicted all 49 RNA editing sites (Table 1) in 23 genes of *D. nobile* from at least 75% of the reference organisms (*Nicotiana tabacum*, *Oryza sativa Japonica* Group, *Phalaenopsis aphrodite* subsp. *Formosana*, *Physcomitrella patens* subsp. Patens and *Zea mays*) and resulted in amino acid substitutions. All the RNA-editing sites were non-silent and edited C to U. Of the 49 RNA editing sites 89.8% (44) editing sites appeared in the second position of triplet codon, 10.2% (five) editing sites appeared in the first position of triplet codon whereas no editing sites appeared in the third base of triplet codon. The genes ndhD, rpoB, rpoC1 had eight, six, and four RNA editing sites, respectively. All the 49 RNA editing sites led to changes in the amino acid. The most frequent amino acid conversion was hydrophilic to hydrophobic (S to L, 22 occurrences and S to F, eight occurrences), followed by hydrophobic to hydrophobic conversions (P to L, 12 occurrences). Seven conversions were found to be hydrophilic to hydrophilic (H to Y, five occurrences and T to M, two occurrences).

## Comparison with other chloroplast genomes within the genus *Dendrobium*

We compared thirty-four chloroplast genomes representing different species within the genus *Dendrobium* (Table 2). The length of the *Dendrobium* species cp genomes ranged from 148,778 to 153,953 bp, with *D. chrysotoxum* being the largest cp genome and *D. moniliforme* the smallest. The cp genomes have acquired the familial angiosperm plastome organization comprising of a LSC, an SSC and a pair of IR regions each. *Dendrobium* cp genomes are also AT-rich (62.26–62.39%) quite similar to other orchid cp genomes (*Zhitao et al., 2017*). Differences in the cp genome size of these species are primarily due to the variations in the length of LSC, SSC and IR regions. Synteny comparison revealed a lack of genome rearrangement and inversions, thereby, substantiating for the highly conserved nature in the genomic structure, including gene number and gene order in these cp genomes. However, structural variation was predominant in the LSC/IR/SSC boundaries (Fig. 2), which could be harnessed for predicting potential biomarkers for species identification.

IR regions are generally considered to be highly conserved regions in the chloroplast genome. IR expansion or contraction is determined by the variability of genes flanking IR/SC junctions (*Huelsenbeck & Ronquist, 2001*). In the evolutionary ladder, SSC and IR border regions experience expansion and contraction that overall contribute to the variation in chloroplast genome length (*Wang et al., 2008*; *Li et al., 2013*). At the IR/LSC boundaries, most IRs of non-orchid monocots exhibit trnH-rps19 gene clusters, excluding Ψrpl22 genes, leading to more-progressive expansion of IRs compared to non-monocot angiosperms (*Yang et al., 2010*; *Goulding et al., 1996*). Contrarily, the orchid chloroplast genomes have distinct characteristics at the IR/SSC junction and are classified into four types based on the organization of genes flanking the $IR_B$/SSC junction ($J_{SB}$). In type I structure, $J_{SB}$ is located upstream of the *ndh*F-*rpl*32 cluster and is primarily seen in *Cypripedium* and *Dendrobium* species. Type II junction is found in *Cymbidium* species in which $J_{SB}$ is located within Ψ*ycf*1 and *ndh*F genes. Type III is reported in *Oncidium*, *Erycina*, and *Phalaenopsis equestris*, in which $J_{SB}$ is located inside the Ψ*ycf*1-*rpl*32 cluster, with the loss of *ndh*F gene. The type IV structure is characterized by the incorporation

*Peer*J

**Table 1** **RNA editing sites predicted in *Dendrobium nobile* chloroplast genome along with its GO annotations.** *D. nobile* cp genome was BLAST aligned against reference datasets of *Nicotiana tabacum*, *Oryza sativa Japonica* Group, *Phalaenopsis aphrodite subsp. Formosana*, *Physcomitrella patens subsp. Patens* and *Zea mays*. Threshold for congruent prediction of RNA editing sites from the reference taxa was set to ≥3 (Count) and 75% (Percentage of prevalence). Count is in the form of (number of reference taxa against which editing site found)/(number of taxa with the homologous site). Further, the genes were exported to OMIX box, blast aligned and subsequently mapped and annotated with Gene ontology (GO) slim terms. Their corresponding GO ids and annotations are shown in the table.

| Gene | GO IDs | GO slim annotation | Nucleotide position | Amino acid position | Triplet position within codon | Base conversion | Codon change | Amino acid conversion | Count | Percentage of Prevalence |
|---|---|---|---|---|---|---|---|---|---|---|
| matK | F: GO:0005198 | F: structural molecule activity | 1258 | 420 | 1 | C→U | CAC→UAC | H→Y | 4/5 | 80 |
| | P: GO:0006412 | P: translation | 913 | 305 | 1 | C→U | CAU→UAU | H→Y | 4/5 | 80 |
| | C: GO:0005840; GO:0009507 | C: ribosome; chloroplast | | | | | | | | |
| rps16 | F: GO:0000166; GO:0005215 | F: nucleotide binding; transporter activity | 143 | 48 | 2 | C→U | UCA→UUA | S→L | 4/4 | 100 |
| | P: GO:0006139; GO:0006810; GO:0009058 | P: nucleobase-containing compound metabolic process; transport; biosynthetic process; | | | | | | | | |
| | C: GO:0009507; GO:0009579; GO:0016020 | C: chloroplast; thylakoid; membrane | | | | | | | | |
| atpA | F: GO:0000166; GO:0005215 | F: nucleotide binding; transporter activity | 773 | 258 | 2 | C→U | UCA→UUA | S→L | 5/5 | 100 |
| | P: GO:0006139; GO:0006810; GO:0009058 | P: nucleobase-containing compound metabolic process; transport; biosynthetic process | | | | | | | | |
| | C: GO:0009507; GO:0009579; GO:0016020 | C: chloroplast; thylakoid; membrane | | | | | | | | |
| atpF | F: GO:0005215 | F: transporter activity | 92 | 31 | 2 | C→U | CCA→CUA | P→L | 5/5 | 100 |
| | P: GO:0006139; GO:0006810; GO:0009058 | P: nucleobase-containing compound metabolic process; transport; biosynthetic process | | | | | | | | |
| | C: GO:0009507; GO:0009579; GO:0016020 | C: chloroplast; thylakoid; membrane | | | | | | | | |
| atpI | F: GO:0005215 | F: transporter activity | 629 | 210 | 2 | C→U | UCA→UUA | S→L | 5/5 | 100 |
| | P: GO:0006139; GO:0006810; GO:0009058 | P: nucleobase-containing compound metabolic process; transport; biosynthetic process | 428 | 143 | 2 | C→U | CCU→CUU | P→L | 5/5 | 100 |
| | C: GO:0005886; GO:0009507; GO:0009579 | C: plasma membrane; chloroplast; thylakoid | | | | | | | | |

Konhar et al. (2019), *PeerJ*, DOI 10.7717/peerj.7756

**Table 1** (*continued*)

| Gene | GO IDs | GO slim annotation | Nucleotide position | Amino acid position | Triplet position within codon | Base conversion | Codon change | Amino acid conversion | Count | Percentage of Prevalence |
|---|---|---|---|---|---|---|---|---|---|---|
| | F: GO:0003677; GO:0016740 | F: DNA binding; transferase activity | 617 | 206 | 2 | C→U | UCG→UUG | S→L | 5/5 | 100 |
| | P: GO:0006139; GO:0009058 | P: nucleobase-containing compound metabolic process; biosynthetic process | 488 | 163 | 2 | C→U | UCA→UUA | S→L | 5/5 | 100 |
| rpoC1 | | | 182 | 61 | 2 | C→U | UCU→UUU | S→F | 5/5 | 100 |
| | C: GO:0009507 | C: chloroplast | 41 | 14 | 2 | C→U | CCA→CUA | P→L | 5/5 | 100 |
| | F: GO:0003677; GO:0016740 | F: DNA binding; transferase activity | 2426 | 809 | 2 | C→U | UCA→UUA | S→L | 4/5 | 80 |
| | | | 623 | 208 | 2 | C→U | CCG→CUG | P→L | 4/5 | 80 |
| | P: GO:0006139; GO:0009058 | P: nucleobase-containing compound metabolic process; biosynthetic process | 566 | 189 | 2 | C→U | UCG→UUG | S→L | 5/5 | 100 |
| rpoB | | | 551 | 184 | 2 | C→U | UCA→UUA | S→L | 5/5 | 100 |
| | C: GO:0009507 | C: chloroplast | 473 | 158 | 2 | C→U | UCG→UUG | S→L | 5/5 | 100 |
| | | | 338 | 113 | 2 | C→U | UCU→UUU | S→F | 5/5 | 100 |
| | F: GO:0003723; GO:0005198 | F: RNA binding; structural molecule activity | | | | | | | | |
| rps14 | P: GO:0006091; GO:0006412; GO:0015979 | P: generation of precursor metabolites and energy; translation; photosynthesis | 149 | 50 | 2 | C→U | CCA→CUA | P→L | 5/5 | 100 |
| | C: GO:0009507; GO:0009579; GO:0016020; GO:0005840 | C: chloroplast; thylakoid; membrane; ribosome | | | | | | | | |
| | F: GO:0005515 | F: protein binding | 191 | 64 | 2 | C→U | CCA→CUA | P→L | 5/5 | 100 |
| ycf3 | P: GO:0015979 | P: photosynthesis | 185 | 62 | 2 | C→U | ACG→AUG | T→M | 5/5 | 100 |
| | C: GO:0009507; GO:0009579; GO:0016020 | C: chloroplast; thylakoid; membrane | 44 | 15 | 2 | C→U | UCU→UUU | S→F | 5/5 | 100 |
| | F: GO:0000166; GO:0005215 | F: nucleotide binding; transporter activity | | | | | | | | |
| atpB | P: GO:0006139; GO:0006810; GO:0009058 | P: nucleobase-containing compound metabolic process; transport; biosynthetic process | 1184 | 395 | 2 | C→U | UCA→UUA | S→L | 5/5 | 100 |
| | C: GO:0009507; GO:0009579; GO:0016020 | C: chloroplast; thylakoid; membrane | | | | | | | | |
| | F: GO:0000166; GO:0016740 | F: nucleotide binding; transporter activity | 1184 | 395 | 2 | C→U | UCA→UUA | S→L | 4/4 | 100 |

**Table 1** (*continued*)

| Gene | GO IDs | GO slim annotation | Nucleotide position | Amino acid position | Triplet position within codon | Base conversion | Codon change | Amino acid conversion | Count | Percentage of Prevalence |
|---|---|---|---|---|---|---|---|---|---|---|
| | P: GO:0006139; GO:0006629; GO:0009058 | P: nucleobase-containing compound metabolic process; lipid metabolic process; biosynthetic process | 1412 | 471 | 2 | C→U | CCA→CUA | P→L | 3/3 | 100 |
| accD | | | | | | | | | | |
| | C: GO:0009507 | C: chloroplast | 1430 | 477 | 2 | C→U | CCU→CUU | P→L | 3/3 | 100 |
| psaI | P: GO:0015979; | P: photosynthesis | 80 | 27 | 2 | C→U | UCU→UUU | S→F | 5/5 | 100 |
| | C: GO:0009507; GO:0009579; GO:0016020 | C: chloroplast; thylakoid; membrane | | | | | | | | |
| | F: GO:0003824; GO:0005488 | F: catalytic activity; binding | | | | | | | | |
| | P: GO:0006091; GO:0015979; | P: generation of precursor metabolites and energy; photosynthesis | 77 | 26 | 2 | C→U | UCU→UUU | S→F | 5/5 | 100 |
| psbF | | | | | | | | | | |
| | C: GO:0005739; GO:0009507; GO:0009579; GO:0016020 | C: mitochondrion; chloroplast; thylakoid; membrane | | | | | | | | |
| petL | F: GO:0003824 | F: catalytic activity | 5 | 2 | 2 | C→U | CCU→CUU | P→L | 5/5 | 100 |
| | C: GO:0009579 | C: thylakoid | | | | | | | | |
| | F: GO:0003723; GO:0005198 | F: RNA binding; structural molecule activity | | | | | | | | |
| | P: GO:0006412; GO:0016043 | P: translation; cellular component organization | 308 | 103 | 2 | C→U | UCA→UUA | S→L | 4/5 | 80 |
| rpl20 | | | | | | | | | | |
| | C: GO:0005840; GO:0009507 | C: ribosome; chloroplast | | | | | | | | |
| | F: GO:0016787 | F: hydrolase activity | 559 | 187 | 1 | C→U | CAU→UAU | H→Y | 5/5 | 100 |
| clpP | P: GO:0019538 | P: protein metabolic process | 82 | 28 | 1 | C→U | CAU→UAU | H→Y | 5/5 | 100 |
| | C: GO:0009507 | C: chloroplast | | | | | | | | |
| | F: GO:0003824; GO:0005488 | F: catalytic activity; binding | | | | | | | | |
| | P: GO:0006091; GO:0015979 | P: generation of precursor metabolites and energy; photosynthesis | 611 | 204 | 2 | C→U | UCA→UUA | S→L | 5/5 | 100 |
| petB | | | | | | | | | | |
| | C:GO:0009507; GO:0009579; GO:0016020 | C: chloroplast; thylakoid; membrane | | | | | | | | |
| | F: GO:0003677; GO:0005515; GO:0016740 | F: DNA binding; protein binding; transferase activity | 830 | 277 | 2 | C→U | UCA→UUA | S→L | 4/4 | 100 |

Konhar et al. (2019), *PeerJ*, DOI 10.7717/peerj.7756

**Table 1** (*continued*)

| Gene | GO IDs | GO slim annotation | Nucleotide position | Amino acid position | Triplet position within codon | Base conversion | Codon change | Amino acid conversion | Count | Percentage of Prevalence |
|---|---|---|---|---|---|---|---|---|---|---|
| | P: GO:0006139; GO:0009058 | P: nucleobase-containing compound metabolic process; biosynthetic process | 368 | 123 | 2 | C→U | UCA→UUA | S→L | 4/4 | 100 |
| rpoA | C: GO:0009507 | C: chloroplast | 200 | 67 | 2 | C→U | UCU→UUU | S→F | 3/4 | 75 |
| | F: GO:0003723; GO:0005198; GO:0016740 | F: RNA binding; structural molecule activity; transferase activity | | | | | | | | |
| rpl2 | P: GO:0006412 | P: translation | 2 | 1 | 2 | C→U | ACG→AUG | T→M | 5/5 | 100 |
| | C: GO:0005840; GO:0009507 | C: ribosome; chloroplast | | | | | | | | |
| | F: GO:0003824; GO:0005488 | F: catalytic activity; binding | 878 | 293 | 2 | C→U | UCA→UUA | S→L | 4/4 | 100 |
| ndhD | P: GO:0006091 | P: generation of precursor metabolites and energy | 674 | 225 | 2 | C→U | UCG→UUG | S→L | 4/4 | 100 |
| | C: GO:0009507; GO:0009579; GO:0016020 | C: chloroplast; thylakoid; membrane | 383 | 128 | 2 | C→U | UCA→UUA | S→L | 4/4 | 100 |
| | F: GO:0003824; GO:0005488 | F: catalytic activity; binding | | | | | | | | |
| ndhA | P: GO:0006091; GO:0015979 | P: generation of precursor metabolites and energy; photosynthesis | 473 | 158 | 2 | C→U | UCA→UUA | S→L | 4/4 | 100 |
| | C: GO:0005886; GO:0009507; GO:0009579 | C: plasma membrane; chloroplast; thylakoid | | | | | | | | |
| | F: GO:0003824; GO:0005488 | F: catalytic activity; binding | 149 | 50 | 2 | C→U | UCA→UUA | S→L | 4/4 | 100 |
| | | | 467 | 156 | 2 | C→U | CCA→CUA | P→L | 4/4 | 100 |
| | | | 586 | 196 | 1 | C→U | CAU→UAU | H→Y | 4/4 | 100 |
| ndhB | P: GO:0006091; GO:0015979 | P: generation of precursor metabolites and energy; photosynthesis | 704 | 235 | 2 | C→U | UCC→UUC | S→F | 4/4 | 100 |
| | | | 737 | 246 | 2 | C→U | CCA→CUA | P→L | 4/4 | 100 |
| | | | 830 | 277 | 2 | C→U | UCA→UUA | S→L | 4/5 | 80 |
| | C: GO:0005886; GO:0009507; GO:0009579 | C: plasma membrane; chloroplast; thylakoid | 836 | 279 | 2 | C→U | UCA→UUA | S→L | 4/5 | 80 |
| | | | 1481 | 494 | 2 | C→U | CCA→CUA | P→L | 4/4 | 100 |

Konhar et al. (2019), *PeerJ*, DOI 10.7717/peerj.7756

Peerj

**Table 1** (*continued*)

| Gene | GO IDs | GO slim annotation | Nucleotide position | Amino acid position | Triplet position within codon | Base conversion | Codon change | Amino acid conversion | Count | Percentage of Prevalence |
|---|---|---|---|---|---|---|---|---|---|---|
| | F: GO:0003723; GO:0005198 | F: RNA binding; structural molecule activity | | | | | | | | |
| rpl23 | P: GO:0006412 | P: translation | 71 | 24 | 2 | C→U | UCU→UUU | S→F | 4/5 | 80 |
| | C: GO:0005840; GO:0009507 | C: ribosome; chloroplast | | | | | | | | |

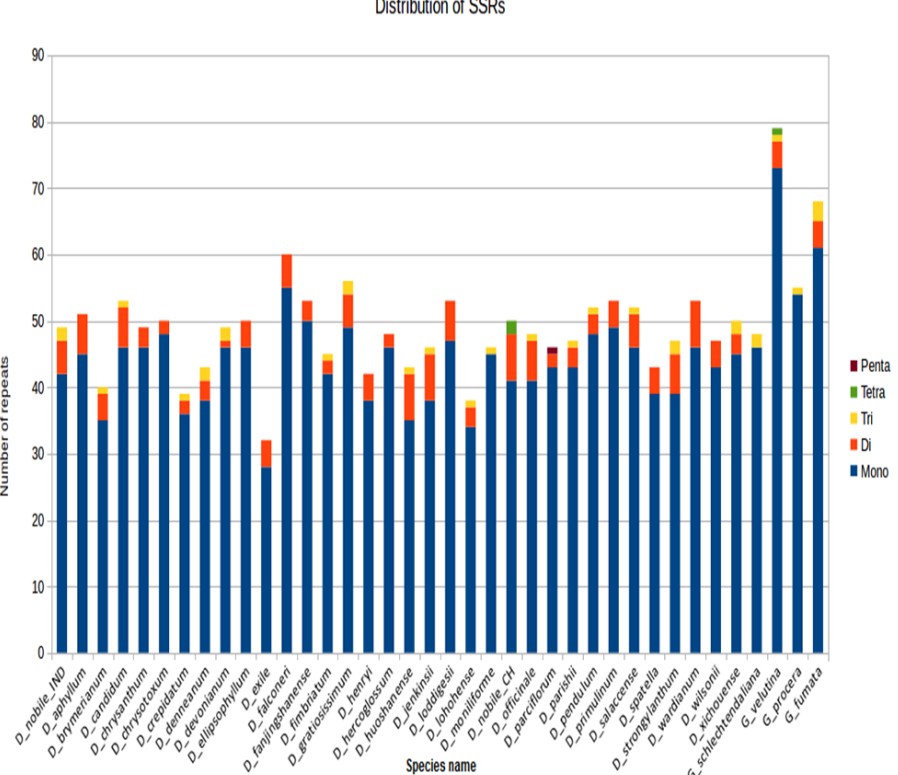

**Figure 4  SSR distribution among different *Dendrobium* plastomes.** The SSR were determined in MISA per scripts based on the comparison between plastomes of each tested *Dendrobium* species and *D. nobile*. Histograms with different color codes indicate the numbers of SSRs. The minimum number (thresholds) of SSRs was set as 10, 5, 4, 3, and 3 for mono-, di-, tri-, tetra-, and penta-nucleotides SSRs, respectively.

of the entire *ycf* 1 into the SSC, with J$_{SB}$ inside *trn*N-*rpl*32 (*Gardner, Slezak & Hall, 2015*). In the present study, the positions of LSC/IRA/SSC/IRB borders were examined in the overall alignment of *Dendrobium* whole cp genomes and all of them were found to have similar structures at the IR/LSC junction akin to type I structure (Fig. 3). Previous studies emphasize that IR expansion or contraction may not correlate with the taxonomic relationships (*Chen & Bundschuh, 2012*). More molecular data is required for enhancing our present understanding of the genes flanking IR/SSC junctions and their underlying variations.

A comparative nucleotide sequence statistics (counts of annotations, AT/GC counts, nucleotide frequency in codon positions etc.) for all the *Dendrobium* species including representatives from outgroup are outlined in Tables 3, 4 and 5. The relative synonymous codon usage is given in parentheses following the codon frequency (averages over all taxa) involved (Table 6). Maximum Likelihood analysis of natural selection codon-by-codon was carried out. For each codon, estimates of the numbers of inferred synonymous (s) and nonsynonymous (n) substitutions are presented along with the number of sites that are estimated to be synonymous (S) and nonsynonymous (N) (Table S1). These estimates were calculated using the joint Maximum Likelihood reconstructions of ancestral states under a

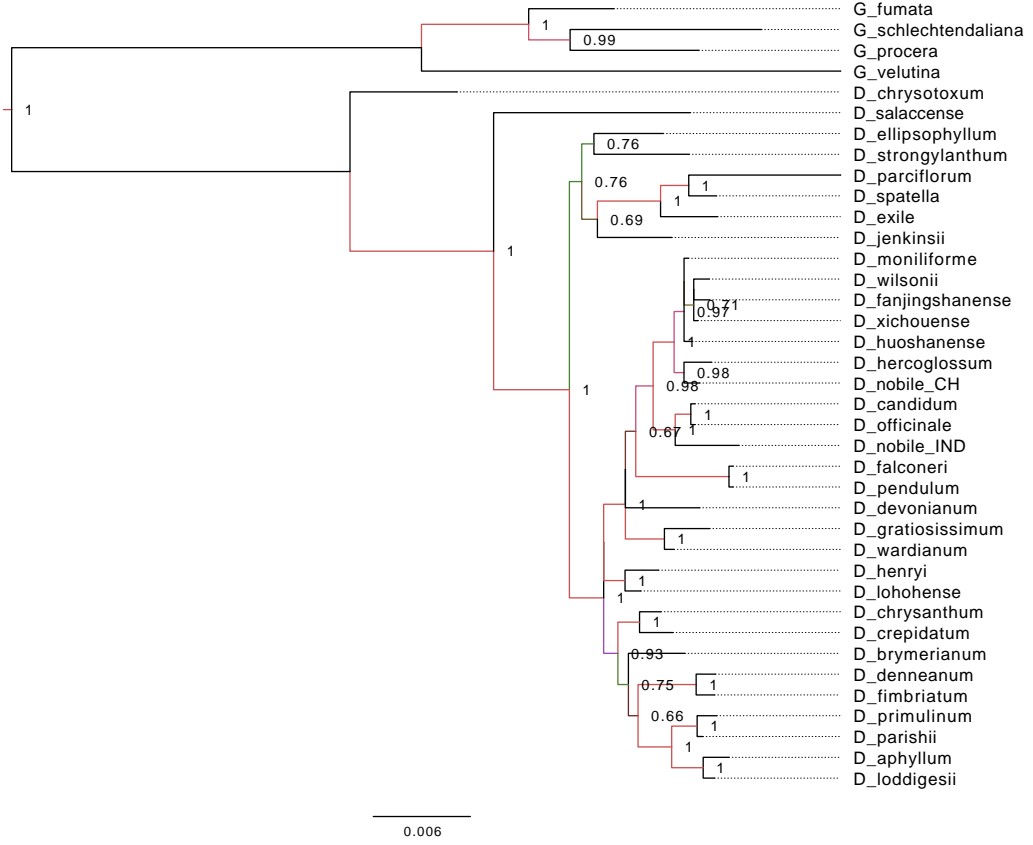

**Figure 5** **Phylogenetic tree based on Bayesian inference from the whole genome alignment matrix of** *Dendrobium* **chloroplast genomes.** The tree yielded monophyletic groupings of the genus *Dendrobium* and *Goodyera* species emerged as outgroup with a separate clade. Posterior probability/bootstrap values are indicated on the internal nodes, which are highly supportive of the overall tree topology.

Muse-Gaut model (*Muse & Gaut, 1994*) of codon substitution and Felsenstein 1981 model (*Felsenstein, 1981*) of nucleotide substitution. For estimating ML values, a tree topology was automatically computed. The test statistic dN-dS was used for detecting codons that have undergone positive selection, where dS is the number of synonymous substitutions per site (s/S) and dN is the number of nonsynonymous substitutions per site (n/N). A positive value for the test statistic indicates an overabundance of nonsynonymous substitutions. In this case, the probability of rejecting the null hypothesis of neutral evolution (*p*-value) was calculated (*Kosakovsky Pond & Frost, 2005*; *Suzuki & Gojobori, 1999*). A value of p less than 0.05 was considered significant at a 5% level and was highlighted (Table S2). Normalized dN-dS for the test statistic is obtained using the total number of substitutions in the tree (measured in expected substitutions per site). The analysis involved 38 nucleotide sequences. Codon positions included were 1st+2nd+3rd+non-coding and all positions containing gaps and missing data were eliminated. There were a total of 108,594 positions in the final dataset.
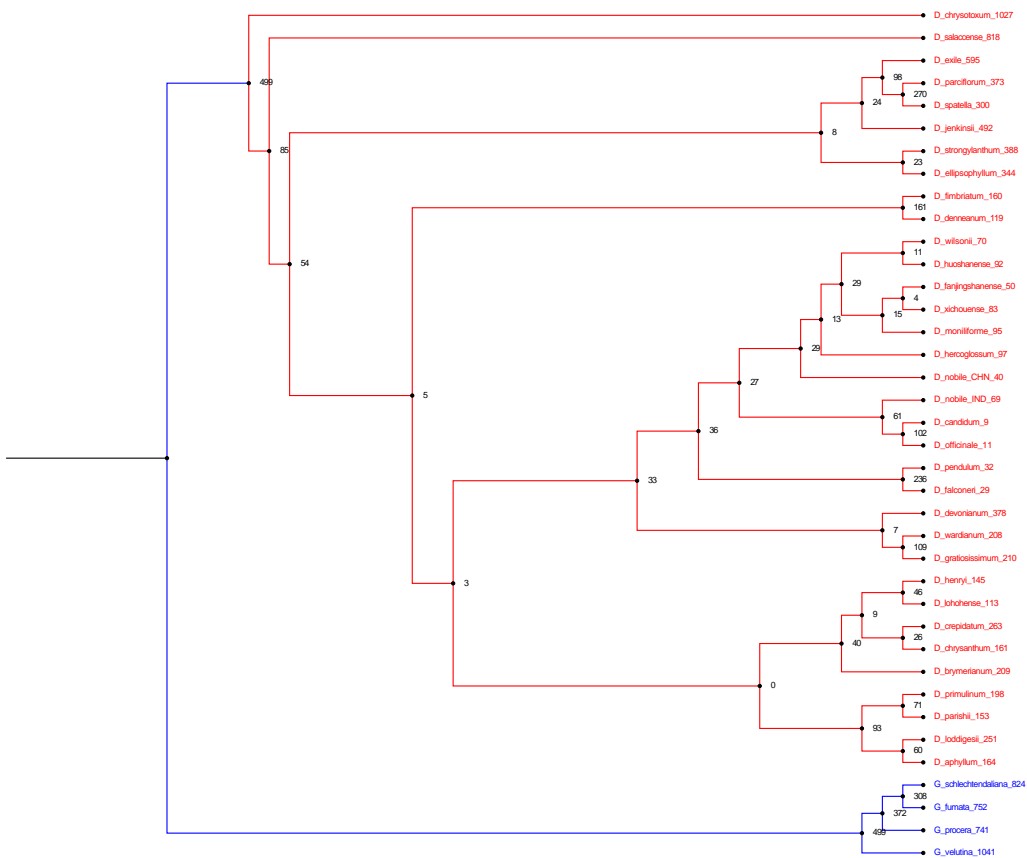

**Figure 6  Alignment free phylogenetic tree reconstruction based on SNP identification.** The optimum kmer size for the dataset is determined that calculates FCK, a measure of diversity of sequences in the dataset (Kchooser) and a consensus of the equally most parsimonious trees are reported. The numbers at the nodes indicate the number of SNPs that are present in all of the descendants of that node and absent in others. The numbers within parentheses at the tips indicate the number of SNPs unique to each particular species.

## Gene ontology analysis

The GO annotation revealed majority of the chloroplast genes are involved in the process of translation, photosynthesis, ion transport and transcription (Table 1). The molecular functions of the genes are majorly binding—RNA, metal ion, DNA, ion and electron transport, RNA polymerase activity and various other enzymatic activities. Enzyme classification showed seven genes to be translocases, four as transferases, two as oxidoreductases, and one each as hydrolase, lyase and ligase. A majority of the genes encode proteins localizing in chloroplast thylakoid membrane, ribosome and few are transported to the mitochondria. The ndhB gene is involved in photosynthesis, while rpoB and rpoC1 are involved in biosynthetic process.

## Characterization of simple sequence repeats

Previous studies have documented prevalence of mononucleotide and dinucleotide SSRs in atleast 15 *Dendrobium* species from 92 syntenic intergenic and intronic loci. Of all these

**Table 2** Summary of characteristics in chloroplast genome sequences of thirty-four *Dendrobium* species and four *Goodyera* species (taken as outgroup).

| Organism | Accession number | Length | Weight (single-stranded) Mda | Weight (double-stranded) Mda |
|---|---|---|---|---|
| *Dendrobium nobile* | KX377961 | 152,018 | 46.932 | 93.912 |
| *Dendrobium officinale* | NC_024019 | 152,221 | 46.995 | 94.038 |
| *Dendrobium strongylanthum* | NC_027691 | 153,059 | 47.256 | 94.556 |
| *Dendrobium huoshanense* | NC_028430 | 153,188 | 47.294 | 94.635 |
| *Dendrobium chrysotoxum* | NC_028549 | 153,953 | 47.528 | 95.108 |
| *Dendrobium nobile (China)* | NC_029456 | 153,660 | 47.453 | 94.927 |
| *Dendrobium pendulum* | NC_029705 | 153,038 | 47.246 | 94.542 |
| *Dendrobium moniliforme* | NC_035154 | 148,778 | 45.931 | 91.911 |
| *Dendrobium primulinum* | NC_035321 | 150,767 | 46.545 | 93.14 |
| *Dendrobium aphyllum* | NC_035322 | 151,524 | 46.779 | 93.607 |
| *Dendrobium brymerianum* | NC_035323 | 151,830 | 46.873 | 93.796 |
| *Dendrobium denneanum* | NC_035324 | 151,565 | 46.793 | 93.633 |
| *Dendrobium devonianum* | NC_035325 | 151,945 | 46.909 | 93.867 |
| *Dendrobium falconeri* | NC_035326 | 151,890 | 46.891 | 93.833 |
| *Dendrobium gratiosissimum* | NC_035327 | 151,829 | 46.873 | 93.796 |
| *Dendrobium hercoglossum* | NC_035328 | 151,939 | 46.908 | 93.864 |
| *Dendrobium wardianum* | NC_035329 | 151,788 | 46.861 | 93.77 |
| *Dendrobium wilsonii* | NC_035330 | 152,080 | 46.951 | 93.951 |
| *Dendrobium crepidatum* | NC_035331 | 151,717 | 46.837 | 93.726 |
| *Dendrobium salaccense* | NC_035332 | 151,104 | 46.648 | 93.347 |
| *Dendrobium spatella* | NC_035333 | 151,829 | 46.872 | 93.796 |
| *Dendrobium parciflorum* | NC_035334 | 150,073 | 46.331 | 92.711 |
| *Dendrobium henryi* | NC_035335 | 151,850 | 46.88 | 93.809 |
| *Dendrobium chrysanthum* | NC_035336 | 151,790 | 46.861 | 93.772 |
| *Dendrobium jenkinsii* | NC_035337 | 151,717 | 46.839 | 93.726 |
| *Dendrobium lohohense* | NC_035338 | 151,812 | 46.868 | 93.785 |
| *Dendrobium parishii* | NC_035339 | 151,689 | 46.83 | 93.709 |
| *Dendrobium ellipsophyllum* | NC_035340 | 152,026 | 46.935 | 93.917 |
| *Dendrobium xichouense* | NC_035341 | 152,052 | 46.942 | 93.933 |
| *Dendrobium fimbriatum* | NC_035342 | 151,673 | 46.825 | 93.699 |
| *Dendrobium exile* | NC_035343 | 151,294 | 46.707 | 93.465 |
| *Dendrobium fanjingshanense* | NC_035344 | 152,108 | 46.96 | 93.968 |
| *Dendrobium candidum* | NC_035745 | 152,094 | 46.955 | 93.959 |
| *Dendrobium loddigesii* | NC_036355 | 152,493 | 47.077 | 94.205 |
| *Goodyera fumata* | NC_026773 | 155,643 | 48.048 | 96.151 |
| *Goodyera procera* | NC_029363 | 153,240 | 47.306 | 94.667 |
| *Goodyera schlechtendaliana* | NC_029364 | 154,348 | 47.648 | 95.351 |
| *Goodyera velutina* | NC_029365 | 152,692 | 47.138 | 94.328 |

**Table 3** Summary features of chloroplast genome sequences of thirty-four *Dendrobium* species and four *Goodyera* species.

| Organism | CDS | Exon | Gene | Misc. feature | Repeat region | rRNA | tRNA |
|---|---|---|---|---|---|---|---|
| *Dendrobium nobile* | 79 | 22 | 132 | 2 | 2 | 8 | 38 |
| *Dendrobium officinale* | 76 | 0 | 129 | 0 | 0 | 8 | 38 |
| *Dendrobium strongylanthum* | 77 | 0 | 130 | 2 | 2 | 8 | 38 |
| *Dendrobium huoshanense* | 76 | 0 | 129 | 2 | 2 | 8 | 38 |
| *Dendrobium chrysotoxum* | 63 | 0 | 116 | 2 | 2 | 8 | 38 |
| *Dendrobium nobile (China)* | 77 | 0 | 130 | 2 | 2 | 8 | 38 |
| *Dendrobium pendulum* | 76 | 0 | 129 | 2 | 2 | 8 | 38 |
| *Dendrobium moniliforme* | 73 | 0 | 129 | 11 | 2 | 8 | 39 |
| *Dendrobium primulinum* | 72 | 0 | 132 | 16 | 2 | 8 | 38 |
| *Dendrobium aphyllum* | 72 | 0 | 132 | 16 | 2 | 8 | 38 |
| *Dendrobium brymerianum* | 72 | 0 | 132 | 16 | 2 | 8 | 38 |
| *Dendrobium denneanum* | 72 | 0 | 132 | 16 | 2 | 8 | 38 |
| *Dendrobium devonianum* | 72 | 0 | 132 | 16 | 2 | 8 | 38 |
| *Dendrobium falconeri* | 72 | 0 | 132 | 16 | 2 | 8 | 38 |
| *Dendrobium gratiosissimum* | 72 | 0 | 132 | 16 | 2 | 8 | 38 |
| *Dendrobium hercoglossum* | 72 | 0 | 132 | 16 | 2 | 8 | 38 |
| *Dendrobium wardianum* | 71 | 0 | 131 | 16 | 2 | 8 | 38 |
| *Dendrobium wilsonii* | 72 | 0 | 132 | 16 | 2 | 8 | 38 |
| *Dendrobium crepidatum* | 72 | 0 | 132 | 16 | 2 | 8 | 38 |
| *Dendrobium salaccense* | 72 | 0 | 132 | 16 | 2 | 8 | 38 |
| *Dendrobium spatella* | 72 | 0 | 132 | 16 | 2 | 8 | 38 |
| *Dendrobium parciflorum* | 72 | 0 | 131 | 16 | 2 | 7 | 38 |
| *Dendrobium henryi* | 72 | 0 | 132 | 16 | 2 | 8 | 38 |
| *Dendrobium chrysanthum* | 72 | 0 | 132 | 16 | 2 | 8 | 38 |
| *Dendrobium jenkinsii* | 72 | 0 | 132 | 16 | 2 | 8 | 38 |
| *Dendrobium lohohense* | 72 | 0 | 132 | 16 | 2 | 8 | 38 |
| *Dendrobium parishii* | 72 | 0 | 132 | 16 | 2 | 8 | 38 |
| *Dendrobium ellipsophyllum* | 72 | 0 | 132 | 16 | 2 | 8 | 38 |
| *Dendrobium xichouense* | 72 | 0 | 132 | 16 | 2 | 8 | 38 |
| *Dendrobium fimbriatum* | 72 | 0 | 132 | 16 | 2 | 8 | 38 |
| *Dendrobium exile* | 72 | 0 | 132 | 16 | 2 | 8 | 38 |
| *Dendrobium fanjingshanense* | 72 | 0 | 132 | 16 | 2 | 8 | 38 |
| *Dendrobium candidum* | 75 | 0 | 128 | 0 | 0 | 8 | 38 |
| *Dendrobium loddigesii* | 68 | 0 | 120 | 9 | 0 | 8 | 39 |
| *Goodyera fumata* | 87 | 0 | 133 | 0 | 0 | 8 | 38 |
| *Goodyera procera* | 80 | 0 | 127 | 0 | 0 | 8 | 39 |
| *Goodyera schlechtendaliana* | 81 | 0 | 129 | 0 | 0 | 8 | 40 |
| *Goodyera velutina* | 79 | 0 | 126 | 0 | 0 | 8 | 39 |

loci, 10(mutational hotspots: *psbB-psbT*, *rpl16-rps3*, *trnR-atpA*, *trnL* intron *ndhF-rpl32*, *rpl32-trnL*, *trnT-trnL*, *clpB-psbB*, *rps16-trnQ* and *trnE-trnT*) are reported to be the fastest evolving and are termed as top ten hotspots (*Chen & Bundschuh, 2012*). The SSRs lying in

| Table 4 | Counts of nucleotides in the chloroplast genomes. | | | | | |
|---|---|---|---|---|---|---|
| Nucleotide | Adenine (A) | Cytosine (C) | Guanine (G) | Thymine (T) | C + G | A + T |
| *Dendrobium nobile* | 46576 | 28853 | 28039 | 48381 | 56892 | 94957 |
| *Dendrobium officinale* | 46743 | 28924 | 28107 | 48447 | 57031 | 95190 |
| *Dendrobium strongylanthum* | 46940 | 29147 | 28431 | 48541 | 57578 | 95481 |
| *Dendrobium huoshanense* | 47032 | 29111 | 28316 | 48729 | 57427 | 95761 |
| *Dendrobium chrysotoxum* | 47180 | 29400 | 28492 | 48881 | 57892 | 96061 |
| *Dendrobium nobile (China)* | 47118 | 28871 | 28748 | 48923 | 57619 | 96041 |
| *Dendrobium pendulum* | 46997 | 29122 | 28242 | 48677 | 57364 | 95674 |
| *Dendrobium moniliforme* | 45551 | 28339 | 27520 | 47368 | 55859 | 92919 |
| *Dendrobium primulinum* | 46191 | 28750 | 27909 | 47917 | 56659 | 94108 |
| *Dendrobium aphyllum* | 46417 | 28917 | 28057 | 48133 | 56974 | 94550 |
| *Dendrobium brymerianum* | 46509 | 28968 | 28123 | 48230 | 57091 | 94739 |
| *Dendrobium denneanum* | 46440 | 28913 | 28115 | 48097 | 57028 | 94537 |
| *Dendrobium devonianum* | 46615 | 28943 | 28108 | 48279 | 57051 | 94894 |
| *Dendrobium falconeri* | 46591 | 28911 | 28040 | 48348 | 56951 | 94939 |
| *Dendrobium gratiosissimum* | 46521 | 28954 | 28095 | 48259 | 57049 | 94780 |
| *Dendrobium hercoglossum* | 46592 | 28941 | 28131 | 48275 | 57072 | 94867 |
| *Dendrobium wardianum* | 46479 | 28955 | 28118 | 48236 | 57073 | 94715 |
| *Dendrobium wilsonii* | 46668 | 28948 | 28101 | 48363 | 57049 | 95031 |
| *Dendrobium crepidatum* | 46482 | 28951 | 28056 | 48228 | 57007 | 94710 |
| *Dendrobium salaccense* | 46493 | 28635 | 27735 | 48241 | 56370 | 94734 |
| *Dendrobium spatella* | 46524 | 28969 | 28091 | 48245 | 57060 | 94769 |
| *Dendrobium parciflorum* | 45941 | 28699 | 27829 | 47604 | 56528 | 93545 |
| *Dendrobium henryi* | 46550 | 28936 | 28093 | 48271 | 57029 | 94821 |
| *Dendrobium chrysanthum* | 46519 | 28939 | 28078 | 48254 | 57017 | 94773 |
| *Dendrobium jenkinsii* | 46497 | 28942 | 28105 | 48173 | 57047 | 94670 |
| *Dendrobium lohohense* | 46558 | 28928 | 28098 | 48228 | 57026 | 94786 |
| *Dendrobium parishii* | 46487 | 28924 | 28079 | 48199 | 57003 | 94686 |
| *Dendrobium ellipsophyllum* | 46690 | 28922 | 28091 | 48323 | 57013 | 95013 |
| *Dendrobium xichouense* | 46672 | 28937 | 28098 | 48345 | 57035 | 95017 |
| *Dendrobium fimbriatum* | 46483 | 28932 | 28094 | 48164 | 57026 | 94647 |
| *Dendrobium exile* | 46251 | 28937 | 28065 | 48041 | 57002 | 94292 |
| *Dendrobium fanjingshanense* | 46694 | 28947 | 28115 | 48352 | 57062 | 95046 |
| *Dendrobium candidum* | 46695 | 28914 | 28091 | 48394 | 57005 | 95089 |
| *Dendrobium loddigesii* | 46868 | 28934 | 28064 | 48627 | 56998 | 95495 |
| *Goodyera fumata* | 48186 | 29569 | 28447 | 49441 | 58016 | 97627 |
| *Goodyera procera* | 47095 | 29370 | 28303 | 48472 | 57673 | 95567 |
| *Goodyera schlechtendaliana* | 47822 | 29206 | 28146 | 49174 | 57352 | 96996 |
| *Goodyera velutina* | 47554 | 28694 | 27658 | 48786 | 56352 | 96340 |
**Table 5  Counts of nucleotide frequency in codon positions across the chloroplast genomes.**

| Nucleotide per position | 1 A | 1 C | 1 G | 1 T | 2 A | 2 C | 2 G | 2 T | 3 A | 3 C | 3 G | 3 T |
|---|---|---|---|---|---|---|---|---|---|---|---|---|
| D. nobile | 0.31 | 0.19 | 0.27 | 0.23 | 0.3 | 0.2 | 0.18 | 0.32 | 0.32 | 0.14 | 0.16 | 0.38 |
| D. officinale | 0.31 | 0.19 | 0.27 | 0.23 | 0.3 | 0.2 | 0.18 | 0.32 | 0.32 | 0.14 | 0.16 | 0.38 |
| D. strongylanthum | 0.31 | 0.19 | 0.27 | 0.23 | 0.3 | 0.2 | 0.18 | 0.32 | 0.32 | 0.14 | 0.16 | 0.38 |
| D. huoshanense | 0.31 | 0.19 | 0.27 | 0.23 | 0.3 | 0.2 | 0.18 | 0.32 | 0.32 | 0.14 | 0.16 | 0.38 |
| D. chrysotoxum | 0.3 | 0.19 | 0.28 | 0.22 | 0.29 | 0.2 | 0.18 | 0.32 | 0.32 | 0.14 | 0.16 | 0.38 |
| D. nobile (China) | 0.31 | 0.19 | 0.27 | 0.23 | 0.3 | 0.2 | 0.18 | 0.32 | 0.32 | 0.14 | 0.16 | 0.38 |
| D. pendulum | 0.31 | 0.19 | 0.27 | 0.23 | 0.3 | 0.2 | 0.18 | 0.32 | 0.32 | 0.14 | 0.16 | 0.38 |
| D. moniliforme | 0.31 | 0.19 | 0.27 | 0.23 | 0.3 | 0.2 | 0.18 | 0.32 | 0.32 | 0.14 | 0.17 | 0.38 |
| D. primulinum | 0.31 | 0.19 | 0.27 | 0.23 | 0.3 | 0.2 | 0.18 | 0.32 | 0.32 | 0.14 | 0.16 | 0.38 |
| D. aphyllum | 0.31 | 0.19 | 0.27 | 0.23 | 0.3 | 0.2 | 0.18 | 0.32 | 0.32 | 0.14 | 0.16 | 0.38 |
| D. brymerianum | 0.31 | 0.19 | 0.27 | 0.23 | 0.3 | 0.2 | 0.18 | 0.32 | 0.32 | 0.14 | 0.16 | 0.38 |
| D. denneanum | 0.31 | 0.19 | 0.27 | 0.23 | 0.3 | 0.2 | 0.18 | 0.32 | 0.32 | 0.14 | 0.16 | 0.38 |
| D. devonianum | 0.31 | 0.19 | 0.27 | 0.23 | 0.3 | 0.2 | 0.18 | 0.32 | 0.32 | 0.14 | 0.16 | 0.38 |
| D. falconeri | 0.31 | 0.19 | 0.27 | 0.23 | 0.3 | 0.2 | 0.18 | 0.32 | 0.32 | 0.14 | 0.16 | 0.38 |
| D. gratiosissimum | 0.31 | 0.19 | 0.27 | 0.23 | 0.3 | 0.2 | 0.18 | 0.32 | 0.32 | 0.14 | 0.17 | 0.38 |
| D. hercoglossum | 0.31 | 0.19 | 0.27 | 0.23 | 0.3 | 0.2 | 0.18 | 0.32 | 0.32 | 0.14 | 0.16 | 0.38 |
| D. wardianum | 0.31 | 0.19 | 0.27 | 0.23 | 0.3 | 0.2 | 0.18 | 0.32 | 0.32 | 0.14 | 0.16 | 0.38 |
| D. wilsonii | 0.31 | 0.19 | 0.27 | 0.23 | 0.3 | 0.2 | 0.18 | 0.32 | 0.32 | 0.14 | 0.16 | 0.38 |
| D. crepidatum | 0.31 | 0.19 | 0.27 | 0.23 | 0.3 | 0.2 | 0.18 | 0.32 | 0.32 | 0.14 | 0.16 | 0.38 |
| D. salaccense | 0.31 | 0.19 | 0.27 | 0.23 | 0.3 | 0.2 | 0.18 | 0.32 | 0.32 | 0.14 | 0.16 | 0.38 |
| D. spatella | 0.31 | 0.19 | 0.27 | 0.23 | 0.3 | 0.2 | 0.18 | 0.32 | 0.31 | 0.14 | 0.17 | 0.38 |
| D. parciflorum | 0.31 | 0.19 | 0.27 | 0.23 | 0.3 | 0.2 | 0.18 | 0.32 | 0.31 | 0.14 | 0.17 | 0.38 |
| D. henryi | 0.31 | 0.19 | 0.27 | 0.23 | 0.3 | 0.2 | 0.18 | 0.32 | 0.32 | 0.14 | 0.16 | 0.38 |
| D. chrysanthum | 0.31 | 0.19 | 0.27 | 0.23 | 0.3 | 0.2 | 0.18 | 0.32 | 0.32 | 0.14 | 0.16 | 0.38 |
| D. jenkinsii | 0.31 | 0.19 | 0.27 | 0.23 | 0.3 | 0.2 | 0.18 | 0.32 | 0.32 | 0.14 | 0.16 | 0.38 |
| D. lohohense | 0.31 | 0.19 | 0.27 | 0.23 | 0.3 | 0.2 | 0.18 | 0.32 | 0.32 | 0.14 | 0.16 | 0.38 |
| D. parishii | 0.31 | 0.19 | 0.27 | 0.23 | 0.3 | 0.2 | 0.18 | 0.32 | 0.32 | 0.14 | 0.17 | 0.38 |
| D. ellipsophyllum | 0.31 | 0.19 | 0.27 | 0.23 | 0.3 | 0.2 | 0.18 | 0.32 | 0.32 | 0.14 | 0.16 | 0.38 |
| D. xichouense | 0.31 | 0.19 | 0.27 | 0.23 | 0.3 | 0.2 | 0.18 | 0.32 | 0.32 | 0.14 | 0.16 | 0.38 |
| D. fimbriatum | 0.31 | 0.19 | 0.27 | 0.23 | 0.3 | 0.2 | 0.18 | 0.32 | 0.32 | 0.14 | 0.16 | 0.38 |
| D. exile | 0.31 | 0.19 | 0.27 | 0.23 | 0.3 | 0.2 | 0.18 | 0.32 | 0.31 | 0.14 | 0.16 | 0.38 |
| D. fanjingshanense | 0.31 | 0.19 | 0.27 | 0.23 | 0.3 | 0.2 | 0.18 | 0.32 | 0.32 | 0.14 | 0.16 | 0.38 |
| D. candidum | 0.31 | 0.19 | 0.27 | 0.23 | 0.3 | 0.2 | 0.18 | 0.32 | 0.32 | 0.14 | 0.16 | 0.38 |
| D. loddigesii | 0.31 | 0.19 | 0.27 | 0.23 | 0.3 | 0.2 | 0.18 | 0.32 | 0.32 | 0.14 | 0.16 | 0.38 |
| G. fumata | 0.31 | 0.19 | 0.26 | 0.24 | 0.29 | 0.2 | 0.18 | 0.33 | 0.32 | 0.14 | 0.16 | 0.38 |
| G. procera | 0.31 | 0.19 | 0.26 | 0.24 | 0.3 | 0.2 | 0.17 | 0.33 | 0.32 | 0.14 | 0.16 | 0.38 |
| G. schlechtendaliana | 0.31 | 0.19 | 0.26 | 0.24 | 0.29 | 0.21 | 0.17 | 0.33 | 0.31 | 0.14 | 0.16 | 0.38 |
| G. velutina | 0.31 | 0.19 | 0.27 | 0.24 | 0.29 | 0.21 | 0.18 | 0.33 | 0.32 | 0.14 | 0.16 | 0.38 |

**Table 6  Relative synonymous codon usage (in parentheses) following the codon frequency across the chloroplast genomes in the genus *Dendrobium*.**

| Codon | Count | RSCU | Codon | Count | RSCU | Codon | Count | RSCU | Codon | Count | RSCU |
|---|---|---|---|---|---|---|---|---|---|---|---|
| UUU(F) | 2018.1 | 1.16 | UCU(S) | 1330 | 1.63 | UAU(Y) | 1371 | 1.38 | UGU(C) | 706.9 | 1.24 |
| UUC(F) | 1459.2 | 0.84 | UCC(S) | 882.8 | 1.08 | UAC(Y) | 621.4 | 0.62 | UGC(C) | 437 | 0.76 |
| UUA(L) | 918.4 | 1.14 | UCA(S) | 999.4 | 1.23 | UAA(*) | 970.5 | 1.05 | UGA(*) | 1065 | 1.15 |
| UUG(L) | 970.9 | 1.21 | UCG(S) | 576.9 | 0.71 | UAG(*) | 732.2 | 0.79 | UGG(W) | 691.4 | 1 |
| CUU(L) | 1068.9 | 1.33 | CCU(P) | 638 | 1.13 | CAU(H) | 919.7 | 1.43 | CGU(R) | 336.1 | 0.63 |
| CUC(L) | 629.2 | 0.78 | CCC(P) | 547.8 | 0.97 | CAC(H) | 369.3 | 0.57 | CGC(R) | 220.7 | 0.41 |
| CUA(L) | 762.8 | 0.95 | CCA(P) | 689.4 | 1.23 | CAA(Q) | 952.8 | 1.38 | CGA(R) | 545.2 | 1.02 |
| CUG(L) | 473.7 | 0.59 | CCG(P) | 375.4 | 0.67 | CAG(Q) | 423.2 | 0.62 | CGG(R) | 343 | 0.64 |
| AUU(I) | 1635.7 | 1.21 | ACU(T) | 646 | 1.21 | AAU(N) | 1580 | 1.39 | AGU(S) | 659.9 | 0.81 |
| AUC(I) | 1072.9 | 0.8 | ACC(T) | 530.8 | 1 | AAC(N) | 695 | 0.61 | AGC(S) | 435.8 | 0.54 |
| AUA(I) | 1337.4 | 0.99 | ACA(T) | 610.3 | 1.15 | AAA(K) | 1914 | 1.31 | AGA(R) | 1171 | 2.2 |
| AUG(M) | 891.4 | 1 | ACG(T) | 343.2 | 0.64 | AAG(K) | 1009 | 0.69 | AGG(R) | 576 | 1.08 |
| GUU(V) | 709.4 | 1.36 | GCU(A) | 467.5 | 1.29 | GAU(D) | 1038 | 1.43 | GGU(G) | 523.7 | 0.99 |
| GUC(V) | 366.7 | 0.7 | GCC(A) | 326.4 | 0.9 | GAC(D) | 413.9 | 0.57 | GGC(G) | 314.4 | 0.59 |
| GUA(V) | 647.8 | 1.24 | GCA(A) | 438.7 | 1.21 | GAA(E) | 1335 | 1.37 | GGA(G) | 754.1 | 1.43 |
| GUG(V) | 366.9 | 0.7 | GCG(A) | 221.5 | 0.61 | GAG(E) | 618.3 | 0.63 | GGG(G) | 521.8 | 0.99 |

these regions could be further investigated for identifying potential markers that can aid in barcoding analysis.

## Phylogenetic analyses

In the present study, we employed two different approaches for phylogeny reconstruction. First we aligned the whole cp genomes and exported the alignment matrices for creating a Bayesian tree (Fig. 5). Two independent MCMC chains were run with first 25% of the cycles removed as burn-in, coalescence of substitution rate and rate model parameters were also examined and average standard deviation of split frequencies was carried out and generations added until the standard deviation value was lowered to 0.01. Secondly we performed a phylogenetic tree construction using an alignment free approach. In this case we identified the SNPs from the cp genomes and utilised them in constructing the phylogenetic tree (Fig. 6). A total of 13,839 SNPs were identified in the 38 genomes analyzed, of which 2,203 were homoplastic SNPs i.e., SNPs that do not correspond to any node in the parsimony tree. The fraction of k-mers present in all genomes is 0.482. The numbers at the nodes in the phylogenetic tree indicate the number of SNPs that are present in all of the descendants of that node and absent in others (Fig. 6). The numbers at the tips indicate the number of SNPs unique to each particular species.

The two different methods that employed both alignment and alignment-free approach resulted in highly reliable identical phylogenetic trees within each data set. Different analyses based on the two datasets generated largely congruent topologies (Figs. 5 and 6) with *Dendrobium* species forming one clade and *Goodyera* species forming another clade as an outgroup.

## CONCLUSIONS

This study provides the first comparative account on the complete chloroplast genome of *D. nobile* from north-east India with 33 other species from the genus *Dendrobium* that revealed higher sequence variation in SSC and LSC regions compared with IR regions in both coding and non-coding regions. The gene order, gene content and genomic structure were highly conserved with those of other sequenced *Dendrobium* species. However, IR contraction is observed within the genus and several SNPs identified from these cp genomes were quite instrumental in generating alignment-free robust phylogenetic trees that congrued with trees generated from aligned matrices of whole cp genomes. This gives an indication that the SNPs, insertions and deletions, LSC and SSC regions in the cp genomes of this medicinal orchid genus can be utilized for barcoding and biodiversity studies. Further, this would augment more and more plastome sequencing of *Dendrobium* species that are not reported in this study.

### Funding

This work was funded under the (Grant ID BT/325/NE/TBP/2012 dated August 07, 2014) by the Department of Biotechnology (DBT), Government of India. The funders had no role in study design, data collection and analysis, decision to publish, or preparation of the manuscript.

### Grant Disclosures

The following grant information was disclosed by the authors:
DBT-NER Twinning program titled ''Next Generation Sequencing (NGS)-based de novo assembly of expressed transcripts and genome information of Orchids in North-East India'': ID BT/325/NE/TBP/2012.

### Competing Interests

The authors declare there are no competing interests.

### Author Contributions

- Ruchishree Konhar conceived and designed the experiments, performed the experiments, analyzed the data, prepared figures and/or tables, authored or reviewed drafts of the paper, approved the final draft.
- Manish Debnath performed the experiments, analyzed the data, prepared figures and/or tables, approved the final draft.
- Santosh Vishwakarma analyzed the data, prepared figures and/or tables, approved the final draft.
- Atanu Bhattacharjee contributed reagents/materials/analysis tools, approved the final draft.
- Durai Sundar analyzed the data, contributed reagents/materials/analysis tools, prepared figures and/or tables, approved the final draft.

- Pramod Tandon conceived and designed the experiments, contributed reagents/materials/analysis tools, approved the final draft.
- Debasis Dash conceived and designed the experiments, contributed reagents/materials/analysis tools, authored or reviewed drafts of the paper, approved the final draft.
- Devendra Kumar Biswal conceived and designed the experiments, performed the experiments, analyzed the data, contributed reagents/materials/analysis tools, prepared figures and/or tables, authored or reviewed drafts of the paper, approved the final draft.

### Data Availability

Data is available at NCBI via GenBank accession number KX377961, BioSample accession number SAMN05190527, SRA accession number SRS1473719, BioProject accession number PRJNA323854 and ID 323854.

### Supplemental Information

Supplemental information for this article can be found online at http://dx.doi.org/10.7717/peerj.7756#supplemental-information.

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
