# Peer review of "The complete chloroplast genome of Dendrobium nobile, an endangered medicinal orchid from north-east India and its comparison with related Dendrobium species"

_PeerJ, doi:10.7717/peerj.7756_

## Round 0.1 · original submission · Major Revisions

Sorry for this decision mail: the first round of reviews split to two extremes and so I needed to invite additional reviewers. As you will see from their comments below, they are still split but I think that you have enough supports for proceeding to the revision step. Please read these comments carefully and revise the manuscript wherever you think reasonable; otherwise, state the reason clearly. Looking forward to your revised manuscript.

·

Basic reporting

This is a typical paper of NGS where chloroplast genome of Dendrobium nobile has been sequenced using standard procedure. Authors compare this genome with related chloroplast genomes of Dendrobium species.

Experimental design

Fresh leaves of D. nobile were collected from plants growing in greenhouse of National Research Centre for Orchids, Sikkim, India. Sequenced using Illumina HiSeq 2500-PE150. Standard procedure was used for checking quality of raw data and for assembling of genome. Genome annotation and codon usage was computed using existing tools that include similarity tools like BLAST.

Validity of the findings

Newly sequenced genome was map on related genomes using reference genomes. Different phylogentic trees constructec to show closeness with other related genomes.

Additional comments

Overlall it is important study for researchers working on Dendrobium species. Authors had implement all standard tools to provide comprehensive annotation of newly sequenced genome.

Reviewer 2 ·

Basic reporting

I strongly recommended that the manuscript should be thoroughly checked for language and grammar.

Experimental design

1. The authors should make sure which DNA were they extract, chloroplast DNA or genomic DNA? If the authors use genomic DNA for the sequencing and assemble analysis, “plastid genome (plastome)” may be more suitable to instead of “chloroplast genome”. If the authors use chloroplast DNA for the analysis, they should add the chloroplast extract method in the “Materials and Methods part”.
2. The comparison studies of Dendrobium plastome, i.e., plastomic structure variation, mutational biases, hotspots and polymorphic SSRs selection, phylogeny, have been investigated already; however, none of these references was listed in this study. This manuscript was focus on the comparison analysis of chloroplast genomes of Dendrobium species; the authors should compare their results with those studies, and discuss their difference, find some new insights.
3. RNA editing sites are important for the function of chloroplast proteins, however, only predicted them is meaningless. Those predicted RNA editing sites should be verified by RT-PCR.

Validity of the findings

4. There are two questions in the phylogenetic analysis.
a. The genus of Goodyera belongs to the subfamily of Orchidoideae, while the genus of Dendrobium belongs to the subfamily of Epidendroideae. Why the authors choose the plastomes of Goodyera spp. as outgroups? Why not choose the plastomes in the subfamily of Epidendroideae?
b. There are many complex in the genus of Dendrobium, for example, D. officinale complex and D. moniliforme complex. Only choose one plastome for each species can not reflect their relationships, the author should increase their taxon sampling. Moreover, the authors should use the correct species name. For example, D. candidum and D. officinale.

Additional comments

The manuscript entitled “The complete chloroplast genome of Dendrobium nobile, an endangered medicinal orchid from Northeast India and its comparison with related chloroplast genomes of Dendrobium species” reported the chloroplast genome of Dendrobium nobile and compared it with other genomes of related Dendrobium species. Though their data may provide some insights into the understanding of the evolution patter of Dendrobium chloroplast genome, the present manuscript is not logically assembled, which simply shows the data without comprehensive analysis.

Reviewer 3 ·

Basic reporting

This is a generally well-written paper although some sentences could still be improved. For instance, the title states: "The complete chloroplast genome of Dendrobium nobile, an endangered medicinal orchid from Northeast India and its comparison with related chloroplast genomes of Dendrobium species". Here, the repetition of chloroplast genomes (x2 in the same sentence) is redundant. These typos occur often in the paper but can easily be fixed.

The aims of the paper are clear but the Introduction (as well as the Discussion) sometimes fails to contextualize the results for a wider audience. For instance, several important references are missing to understand the evolution of orchids, and the influence of polyploidy on their diversification:
https://onlinelibrary.wiley.com/doi/abs/10.1111/cla.12153
https://onlinelibrary.wiley.com/doi/abs/10.1111/jbi.12854
https://bmcevolbiol.biomedcentral.com/articles/10.1186/1471-2148-14-20
https://bsapubs.onlinelibrary.wiley.com/doi/abs/10.1002/ajb2.1178

Also: I think the largest orchid genus is Epidendrum (not Dendrobium). Please clarify that.

Experimental design

Research questions are well defined, relevant and meaningful to understand the evolution of Dendrobium. Methods are generally described with sufficient detail except in:

1) Please clarify if some of the analyzed species are polyploid and how you have addressed this in the analysis.
2) A comparison of homologous genes were performed with other chloroplast genomes. Which ones?
3) The gene map on Figure 1 is hard to follow. For instance, the coding genes are not annotated on the gene map of Dendrobium nobile. Why is that?

Validity of the findings

Data is statistically sound and relevant for the genus, and probably also for orchids in general if the authors place the results in a wider context.

Additional comments

I think this is a fine study case that can potentially enlarge our knowledge concerning the evolution of an important genus of orchids: Dendrobium. Conclusions are well stated, linked to original research question and limited to supporting results. They can however be improved if authors backup their conclusions with other studies that have been published in the same subject.

Please clarify how this study contributes to the conservation of Dendrobium nobile. This seems an important question and it is mentioned in the title and in the Introduction but findings are quite unspecific, eg., it is hard to see how this data can contribute to the conservation of this species in its current stage.

In any case, I think these are all minor details that the authors can fix quite easily. So, I hope to see this manuscript published very soon.
Congratulations to the authors.

Reviewer 4 ·

Basic reporting

The manuscript is well organized and written. The Introduction gives a comprehensive background to the study. The language used in the manuscript is clear and professional, however I have some minor comments:
1. line 36: Does the wording: “The present-day Dendrobium is the largest genus” means it is the largest genus among the tribe Dendrobieae or is Dendrobium just more species-rich than it was before? Please refine the statement.
2. lines 70-73: The sentence is long and a little bit confusing. Please, check if the wording “one such endangered orchid... that demands...” is grammatically correct. You could also replace the whole sentence with two shorter ones.
3. Check the text for double spaces and double dots (eg.: line 175, 178, 183).
4. Line 196: “ Synteny comparison...” - shouldn’t it be “Synthetic comparison”?
5. Line 247: “do no” replace with “do not”.
6. line 170: The gene rps12 is absent in the Figure 1.
7. Figure 1.: Lack of rps12 gene. LSC and “source Dendrobium nobile” signatures are hardly readable due to arrows crossing the signatures. If there are no gene names given in the figure, I would recommend deleting fragments of it eg.: ycf2 g... (in the green circle).
8. Figure 2., Figure 5: The are no italics in the caption (Dendrobium, Goodyera).
9. Table 1.: Please, explain in the table caption (or in the main text in the Results), what “count” and “percantage” is. For now, it is too laconic.

Experimental design

Materials and Methods is written flawlessly. It is clear, understandable and comprehensive and the methods used in the study are appropriate for the issue.

Validity of the findings

In the chapter Results and Discussion the results are well presented, however there should be more references to similar studies in a discussion part.
1. Line 194: “...quite similar to other orchid cp genomes.” - reference needed.
2. Line 202: “IR regions are generally considered to be highly conserved regions in the chloroplast genome.” – reference needed.
3. The whole discussion, especially in the part “Comparison with other chloroplast genomes with the genus Dendrobium” needs to be discussed with the results from other studies on different genera. What about the structural variation detected in the LSC/IR/SSC boundaries? Was it observed in other genera among Orchidaceae?

Additional comments

The manuscript entitled „The complete chloroplast genome of Dendrobium nobile, an endangered medicinal orchid from Northeast India and its comparison with related chloroplast genomes of Dendrobium species” is an effect of solid and well-planned study, taking an important and timely issue. Identification of endangered species using molecular markers is nowadays important and necessary skill. Therefore, studies broadening our knowledge on DNA sequence variability between closely related species are always welcome.

Reviewer 5 ·

Basic reporting

The whole logical structure was confused and unclear. The authors stated that they have analyzed the 33 newly sequenced chloroplast genomes retrieved from NCBI Refseq database was compared with that of the first complete chloroplast genome of D. nobile. However, actually, only one newly sequenced chloroplast genome was obtained in the material and methods. In the phylogenetic construction analysis, they have included D. nobile cp genomes from India and China along with 32 other Dendrobium cp genomes. How many species was actually included in the current analysis? Some results and discussion were wrong.

The abstract part must be re-written, the mean of whole abstract part was unclear and unlogical, the important results and conclusions were not summarized, the authors must to re-edit it.

The summary of introduction part was not complete, and some important literatures were not cited, the authors must re-edit their manuscript.

The many parts of the results and discussion were only the results, there are no any discussion to the many results. The authors should propose the important questions in their manuscript, rather than introduce the simple results.


Many sentences were wrong and errors, the authors must re-organize it. E.g.,
The medicinal orchid genus Dendrobium belonging to the Orchidaceae family is the largest genus comprising about 800-1500 species. To better illustrate the species status in the genus Dendrobium, a comparative analysis of 33 newly sequenced chloroplast genomes retrieved from NCBI Refseq database was compared with that of the first complete chloroplast genome of D. nobile from north-east India based on next-generation sequencing methods (Illumina HiSeq 2500-PE150).

Experimental design

unclear

Validity of the findings

inaccurate

Additional comments

The whole logical structure was confused and unclear. The authors stated that they have analyzed the 33 newly sequenced chloroplast genomes retrieved from NCBI Refseq database was compared with that of the first complete chloroplast genome of D. nobile. However, actually, only one newly sequenced chloroplast genome was obtained in the material and methods. In the phylogenetic construction analysis, they have included D. nobile cp genomes from India and China along with 32 other Dendrobium cp genomes. How many species was actually included in the current analysis? Some results and discussion were wrong.

The abstract part must be re-written, the mean of whole abstract part was unclear and unlogical, the important results and conclusions were not summarized, the authors must to re-edit it.

The summary of introduction part was not complete, and some important literatures were not cited, the authors must re-edit their manuscript.

The many parts of the results and discussion were only the results, there are no any discussion to the many results. The authors should propose the important questions in their manuscript, rather than introduce the simple results.


Many sentences were wrong and errors, the authors must re-organize it. E.g.,
The medicinal orchid genus Dendrobium belonging to the Orchidaceae family is the largest genus comprising about 800-1500 species. To better illustrate the species status in the genus Dendrobium, a comparative analysis of 33 newly sequenced chloroplast genomes retrieved from NCBI Refseq database was compared with that of the first complete chloroplast genome of D. nobile from north-east India based on next-generation sequencing methods (Illumina HiSeq 2500-PE150).

External reviews were received for this submission. These reviews were used by the Editor when they made their decision, and can be downloaded below.

---

## Round 0.2 · Minor Revisions

Sorry for the delay of this decision mail. Like the situation in its original version, the opinions by the reviewers on this revised manuscript has been split and one reviewer has not given me his/her report yet. Now, we have one recommendation for its acceptance and the other for its rejection. In addition, our Section Editor points out that an additional minor revision should be taken to highlight the needs for comparative genome comparisons, such as annotating the genome with ontology terms. "This can very easily be added to Table 1. Journal manuscripts are often scanned by text-mining software that locates and extracts core data elements, like gene function. Adding standard ontology terms, such as the Gene Ontology (GO, geneontology.org) or others from the OBO foundry (obofoundry.org) can enhance the recognition of your contribution and description. This will also make human curation of literature easier and more accurate. None of this was visible." Therefore, I recommend its another minor revision to the Editor-in-Chief. Please add the GO terms, as suggested above and resubmit the manuscript, again. Thanks, in advance, for your patience!

Reviewer 2 ·

Basic reporting

The manuscript is not logically assembled, especially for the “Result” and “Discussion” parts.

Experimental design

The author did not well resolve the questions we raised, i.e., verified the RNA editing sites, increase the taxon sampling. These are so important questions for this manuscript which need to improved.

Validity of the findings

Only report one chloroplast genome is not enough to publish a scientific paper, they need to do more comprehensive analyses with other published chloroplast genomes. However, they failed to do so. Moreover, they ignored the questions that we have raised.

Additional comments

Through the authors have made effort to improve their manuscript, the present manuscript is not logically assembled. For example, only two section “Genome organization and features”, “Simple sequence repeat identification” were in the Result part, however, other results were written in the Discussion part. The authors need to reorganize their Result and Discussion parts. Also, in the Instruction part, the authors should focus on introducing the significance of “why they need to sequence and analysis the cp genome of Dendrobium nobile” rather than review the progress of molecular marker studies of orchid and Dendrobium species.

Reviewer 4 ·

Basic reporting

The manuscript is well organized and written. The changes made by the Authors significantly improved the quality of the manuscript.

Experimental design

The article meets standards of the journal,

Validity of the findings

no comment

Additional comments

I consider the article a valuable contribution to the area of plant genomics as well as for research considering Dendrobium species. The manuscript is suitable for publication.

---

## Round 0.3 · accepted · Accept

Your manuscript has been re-reviewed by the reviewer who recommended further revision in its previous version. As you can see from his/her comments below, he/she still raises some points but now agrees to accept it. Thus, I am happy to inform you that I will recommend its acceptance. Congratulations!

Reviewer 2 ·

Basic reporting

The article should include sufficient introduction and background to demonstrate how the work fits into the broader field of knowledge. Relevant prior literature should be appropriately referenced.

Experimental design

Original primary research within Aims and Scope of the journal.

Validity of the findings

Conclusions are well stated,Speculation is welcome

Additional comments

After the revision, the manuscript was great improved and can be accept in current form.